*Correspondence*

# Crystal structure of MAGEA4 MHD-RAD18 R6BD reveals a flipped binding mode compared to AlphaFold2 prediction

Karly Forker [1], Matthew C Fleming[2,4], Kenneth H Pearce[2], Cyrus Vaziri[3], Albert A Bowers[2] & Pei Zhou [1]✉

Comment on: S Griffith-Jones et al (March 2024)
See reply: S Griffith-Jones et al

Recently, Griffith-Jones and colleagues investigated the structural basis of RAD18 regulation by MAGEA4 through NMR titration and AlphaFold2 modeling (Jumper et al, 2021) to elucidate the interaction between the RAD6-binding domain (R6BD) of RAD18 and the MAGE homology domain (MHD) of MAGEA4 (Griffith-Jones et al, 2024). In the Alpha-Fold2 model constructed by Griffith-Jones and colleagues, the R6BD region of RAD18 forms two α-helices: the shorter N-terminal helix interacts with the α4-α5 loop of the MAGEA4 MHD, whereas the longer C-terminal helix lays across the groove formed by α4, α5, and α8 of the MAGEA4 MHD, with its C-terminus pointing towards α1 and α3 of MAGEA4. Such a binding mode is purportedly supported by NMR titration and mutagenesis data.

In our effort to develop peptide inhibitors targeting the RAD18-binding interface of the MHD of MAGEA4 (Fleming et al, 2023, Fleming et al, 2022), we determined the crystal structure of the MAGEA4 MHD/RAD18 R6BD complex at 2.58 Å resolution (Fig. 1A; statistics shown in Table EV1). Each unit cell contains two protomers of the MAGEA4 MHD-RAD18 R6BD complex, with similar conformations. Protomer 1 (with the MAGEA4 MHD in chain A and the RAD18 R6BD in chain B) has slightly better quality and is used for analysis here. The electron density of the RAD18 R6BD (chain B) is very well defined in the map

except for the sidechains of K347 and E349 located at N-terminal end of the R6BD helix (omit map shown in Fig. EV1). We found that the RAD18 R6BD forms a single α-helix and binds to a surface valley of the MAGEA4 MHD formed by helices α1, α3, α4, α5, and α8. Numerous interactions between the hydrophobic residues of the MAGEA4 MHD (M161, I162, F163, I205, M212, V239, V291, and V294) and RAD18 R6BD (A365, I364, Y361, A357, L353, and V354) are observed (Fig. 1B). Surprisingly, we found that the RAD18 R6BD peptide binds in the opposite orientation to the AlphaFold2 model presented by Griffith-Jones and colleagues (compare Fig. 1A with Griffith-Jones et al, Fig. 1B).

Because Griffith-Jones and colleagues used a flexible linker strategy to tether the MAGEA4 MHD and the RAD18 R6BD for AlphaFold2 modeling (Jumper et al, 2021), we reasoned that it is possible that AlphaFold-Multimer (preprint: Evans et al, 2021) designed for modeling protein complexes could mitigate such limitation. Therefore, we constructed five models of the MAGEA4 MHD (residues D101-V317) and the RAD18 R6BD (residues H339-G366) using the Colab-Fold implementation (Mirdita et al, 2022) of AlphaFold-Multimer (preprint: Evans et al, 2021). We found that the top five predicted models segregated into two binding clusters (Fig. EV2). Cluster 1 (consisting of models 3–5) closely matches the reported AlphaFold2 model by Griffith-Jones and colleagues, and model 3 from this cluster is shown in Fig. 1C,D for comparison with the crystal structure of the MAGEA4 MHD-RAD18 R6BD complex

(Fig. 1A,B). Cluster 2, consisting of models 1–2, is shifted ~15 Å relative to the cluster 1. Its binding orientation remains similar to that reported by Griffith-Jones and colleagues and is opposite from that observed in the crystal structure (compare Fig. EV2 with Fig. 1A).

Despite the flipped helix orientation, there is considerable overlap of the interfacial residues of the MAGEA4 MHD between the crystal structure and the AlphaFold models (compare Fig. 1B with Fig. 1D in this manuscript and Griffith-Jones et al, Fig. 1D (Griffith-Jones et al, 2024); shared surface residues from MAGEA4 MHD are labeled in blue in Fig. 1B,D). Therefore, although the reported NMR perturbation and mutagenesis data appeared to be supporting the AlphaFold model, the interacting pairs of MAGEA4 and RAD18 residues are completely scrambled in the AlphaFold2 model in comparison with the experimental structure due to the flipped helix orientation of the RAD18 R6BD.

With the remarkable success of the AlphaFold2 prediction yielding highly accurate structural models (Jumper et al, 2021), studies based entirely on AlphaFold2 analysis have gained popularity. It is, however, important to note that AlphaFold2 is fundamentally a statistical method relying on existing experimental structures in the PDB database, and it is not based on biophysical principles. Hence, its prediction can be biased by existing experimental structures. The case presented here is a sober reminder that despite the inspiring "structures" generated by AlphaFold2, they remain as computer models and require rigorous experimental investigation to

[1]Department of Biochemistry, Duke University School of Medicine, Durham, NC 27710, USA. [2]Division of Chemical Biology and Medicinal Chemistry, UNC Eshelman School of Pharmacy, University of North Carolina at Chapel Hill, Chapel Hill, NC 27599, USA. [3]Curriculum in Genetics and Molecular Biology, Department of Pathology and Laboratory Medicine, and Curriculum in Toxicology, University of North Carolina, Chapel Hill, NC 27599, USA. [4]Present address: Nimble Therapeutics, 603 Science Dr, Madison, WI 53711, USA. ✉E-mail: peizhou@biochem.duke.edu

https://doi.org/10.1038/s44318-024-00140-2 | Published online: 21 June 2024

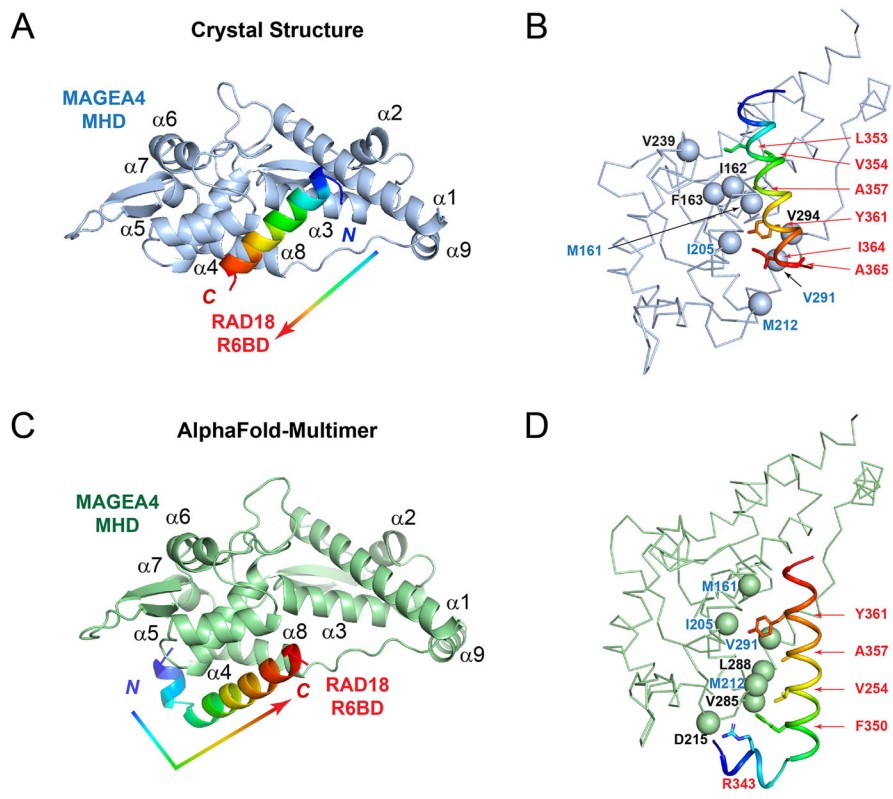

**Figure 1. Comparison of the crystal structure and AlphaFold-Multimer model of the MAGEA4 MHD (D101-V317)-RAD18 R6BD (H339-G366) complex.**

(A) Crystal structure of the MAGEA4-RAD18 R6BD complex in the cartoon representation. The MAGEA4 MHD is colored in light blue, and the RAD18 R6BD is colored in rainbow, with the N-terminus color in blue and C-terminus colored in red. The orientation of the RAD18 R6BD helix is indicated with an arrow. (B) Interface of the MAGEA4 MHD and RAD18 R6BD observed in the crystal structure. Interfacial residues of RAD18 R6BD are shown in the stick model and labeled in red, whereas interfacial residues of MAGEA4 MHD are indicated by Cα spheres and are labeled in black and blue. (C) The AlphaFold-multimer model of the MAGEA4-RAD18 R6BD complex in the cartoon representation. The MAGEA4 MHD is colored in light green, and the RAD18 R6BD is colored in rainbow, with the N-terminus color in blue and C-terminus colored in red. The orientation of the RAD18 R6BD helix is indicated with an arrow. (D) Interface of the MAGEA4 MHD and RAD18 R6BD observed in the crystal structure. Interfacial residues of RAD18 R6BD are shown in the stick model and labeled in red, whereas interfacial residues of MAGEA4 MHD are indicated by Cα spheres and are labeled in black and blue. In panels (B) and (D), shared interfacial residues of MAGEA4 MHD between the crystal structure and the AlphaFold-multimer model are labeled in blue.

unearth the structural basis underlying important biological processes. Our structural elucidation of the MAGEA4 MHD-RAD18 R6BD complex has provided molecular evidence to clarify confusions caused by the AlphaFold2 modeling and accelerate the development of MAGEA4-RAD18 antagonists to dismantle a key enabling feature of cancer cells.

## Limitation of the current study

Despite the striking discrepancy between the AlphaFold2 model and the crystal structure, it is evident that a small peptide of the RAD18 R6BD is not equivalent to full-length RAD18. The binding mode of a small peptide might be affected by crystal packing, and the MAGEA4-RAD18 R6BD complex might exist in solution with an alternative conformation or with multiple conformations. Therefore, the molecular details between full-length MAGEA4 and RAD18 require further investigation.

## Methods

### Protein expression and purification

His₆-TEV-MAGEA4-MHD (residues D101-V317) was expressed in BL21 (DE3) cells using 0.2 mM IPTG induction overnight at 20 °C. Cell pellets were resuspended in a buffer containing 50 mM HEPES pH 7.5, 500 mM NaCl, 10 mM imidazole, 5% glycerol, 0.5 mM TCEP (Tris (2-carboxyethyl) phosphine) and lysed by French Press. Cell debris was removed by centrifugation at 17,500 RPM for 30 min at 4 °C. The supernatant was applied to a 10 mL Ni-NTA gravity flow column equilibrated in the lysis buffer. The column was washed with 5 column volume (CV) of the lysis buffer and batch eluted with 2 CV of buffers (50 mM HEPES pH 7.5, 500 mM NaCl, 0.5 mM TCEP, and 5% glycerol) containing 50, 100, 150, 200, and 250 mM imidazole. MAGEA4-MHD was buffer exchanged 100-fold with the lysis buffer containing 1 mM DTT and then incubated overnight at 4 °C with a 1:20 molar ratio of TEV protease. TEV protease and other contaminants were removed by applying the protein to a 10 mL Ni-NTA gravity flow column. The flow through was collected, buffer exchanged in the size-exclusion chromatography (SEC) buffer consisting of 20 mM Tris HCl (pH 7.5), 200 mM NaCl, 1 mM TCEP, and concentrated to 2 mL. The protein was loaded onto a Superdex-75 gel filtration column pre-equilibrated in the SEC buffer. Fractions containing MAGEA4 were pooled and concentrated.

### RAD18 R6BD

The C-terminally amidated RAD18 R6BD peptide (H339-G366) was synthesized at the University of North Carolina at Chapel Hill (UNC Chapel Hill) High-Throughput Peptide Synthesis and Array Facility. The purity was assessed by HPLC, and the mass verified by MALDI-TOF (Applied Biosystems 4700 Proteomics Analyzer 72102).

### Crystallization of MAGEA4-MHD and RAD18 R6BD

The MAGEA4 MHD (residues 101–317, 10.3 mg/mL) was incubated on ice with a 1:2 molar ratio of RAD18 R6BD (residues 339–366) in 20 mM Tris HCl pH 7.5, 200 mM NaCl, 1 mM TCEP. Microcrystals were grown by sitting drop vapor diffusion with the mixture of 1 µL protein sample and 1 µL mother liquor solution containing 20% (w/v) PEG 3350 and 300 mM ammonium formate. The initial crystals were propagated by streak seeding. Seed stocks were prepared by transferring 2 µL of the crystal-containing drop into a solution containing

its mother liquor. After three days, large rod-shaped crystals were observed in a well containing 20% (w/v) PEG 8000, 100 mM HEPES/NaOH pH 7.5, 200 mM ammonium sulfate, 10% (v/v) 2-propanol. Crystals were cryoprotected using the corresponding mother liquor containing 37.5% glycerol.

**Data collection and model building**

Data sets were collected at the NE-CAT 24-ID-E beamline at the Advanced Photon Source at Argonne National Laboratory. X-ray diffraction data were processed using XDS (Kabsch, 2010), and phasing solved by molecular replacement with the PHASER module in the PHENIX suite (Afonine et al, 2018) using PDB 7UOA as the search model. Iterative model building and refinement were carried out using COOT (Emsley and Cowtan, 2004) and PHENIX (Afonine et al, 2018). The final model has 96.9% residues in the favored region, 3.1% in the allowed region, and 0.0% in the outlier region of the Ramachandran plot.

## Data availability

The structural coordinate from this publication has been deposited to the Protein Data Bank (https://www.rcsb.org/) with the access code of 9BD3.

## Peer review information

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

## Acknowledgements

This work was supported by grants from the National Institutes of Health R01 CA279034 to PZ, R35GM125005 to AB, and R01 CA229530 to CV and KHP. X-ray diffraction data were collected at the Northeastern Collaborative Access Team beamline 24-ID-E, funded by the National Institute of General Medical Sciences (Grant P30 GM124165). This research used resources of the Advanced Photon Source, a US Department of Energy (DOE) Office of Science User Facility operated for the DOE Office of Science by the Argonne National Laboratory under Contract DE-AC02-06CH11357.

## Author contributions

**Karly Forker**: Formal analysis; Validation; Investigation; Writing—original draft; Writing—review and editing. **Matthew C Fleming**: Resources; Investigation; Writing—review and editing. **Kenneth H Pearce**: Resources; Funding acquisition; Writing—review and editing. **Cyrus Vaziri**: Resources; Funding acquisition; Writing—review and editing. **Albert A Bowers**: Conceptualization; Resources; Funding acquisition; Writing—review and editing. **Pei Zhou**: Conceptualization; Resources; Data curation; Formal analysis; Supervision; Funding acquisition; Validation; Investigation; Visualization; Methodology; Writing—original draft; Project administration; Writing—review and editing.

## Disclosure and competing interests statement

The authors declare no competing interests.

# Expanded View Figures

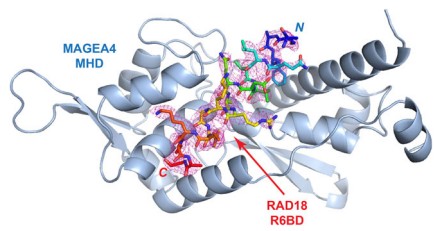

**Figure EV1.  Omit map of the RAD18 R6BD in the MAGEA4 MHD-bound complex.**

Each unit cell contains two protomers with similar conformations. For clarity, only protomer 1 (chain A and chain B) is shown here. The MAGEA4 MHD (chain A) is shown in the cartoon representation and colored in light blue. The RAD18 R6BD (chain B) is shown in the stick model and color in rainbow with the N-terminus in blue and C-terminus in red. The purple mesh represents the 2mFo-DFc omit map of the RAD18 R6BD (chain B) contoured at 1.0 σ. The N-terminus and C-terminus of the RAD18 R6BD are labeled.

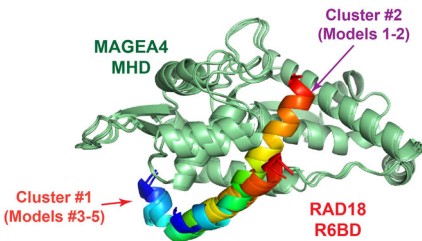

**Figure EV2. Five models of the MAGE MHD-RAD18 R6BD complex.**

The sequences of human MAGEA4 MHD (D101-V317) and RAD18 R6BD (H339-G366) were used as the input for the ColabFold implementation of the AlphaFold-Multimer. The MAGEA4 MHD is colored in pale green and RAD18 R6BD colored in rainbow, with N-terminus in blue and C-terminus in red. The five models of the RAD16 R6BD share similar orientations but are congregated into two clusters, with models #3-#5 into one cluster (cluster #1) that resembles the AlphaFold2 model reported by (Griffith-Jones et al, 2024) and models #1 and #2 into the other cluster (cluster #2).

