## [Peer Review File · The EMBO Journal]

Crystal Structure of MAGEA4 MHD-RAD18 R6BD Reveals a Flipped Binding Mode compared to AlphaFold2 Prediction

Karly Forker, Matthew Fleming, Kenneth Pearce, Cyrus Vaziri, Albert Bowers, and Pei Zhou

Corresponding author(s): Pei Zhou (peizhou@biochem.duke.edu)

Review Timeline:

Submission Date:	18th Apr 24
Editorial Decision:	21st May 24
Revision Received:	22nd May 24
Accepted:	24th May 24

Editor: Hartmut Vodermaier

Transaction Report:

21st May 2024

Re: EMBOJ-2024-117641P

Crystal Structure of the MAGEA4-RAD18 R6BD Complex Reveals a Flipped Binding Mode from the AlphaFold2 Prediction

Dear Dr. Zhou, dear Dr. Bhogaraju,

Thank you for sending in your correspondence and response, respectively, regarding our earlier publication of the article by Griffith-Jones et al., "Structural basis for RAD18 regulation by MAGEA4 and its implications for RING ubiquitin ligase binding by MAGE family proteins". I sent both pieces jointly to referee #2 of the original submission, and to an additional structural expert not involved in the earlier review process. Both of them appreciate the arguments in both communications, and agree with the importance of publishing this exchange and the experimentally-determined structure. As you will see from their feedback copied below, they however suggest a certain amount of rephrasing for both pieces; in particular, the structural referee raises the well-taken point that also a crystal structure by itself does not necessarily reflect the physiological in-solution conformation.

I am therefore inviting you to modify your respective communications in light of these comments, also taking our more specific guidelines below into account, and to send back revised versions at your earliest convenience (ideally, by the end of this week). Please do not hesitate to contact me with any further questions you may have in this regard.

With kind regards,

Hartmut Vodermaier

SPECIFIC GUIDANCE:

- Please upload text and figure as separate, individual files (i.e., one file per figure). Please change Figures S1/2 into Expanded View Figures: calling them Figure EV1/2. Please retain the legends for both main and expanded view figures at the end of the main text. Lastly, please also upload the data collection table separately as "Table EV1", and adjust its callout accordingly.
- When referring to figures in the Griffith-Jones et al paper in the text, please try to differentiate them clearly from references to your own current figures, so that typesetters will not be confused and add inappropriate links
- After the method section, please include a "Data Availability" section (suggested wording: "The structural coordinates from this publication have been deposited to the [name of the database] database [URL] and assigned the identifier [accession #]."); and a Disclosure and Competing Interests Statement (for details, see <https://www.embopress.org/competing-interests>)
- Please adjust the format for citation of preprints as specified in our author guidelines:
The citation in the text should be: "(preprint: NAME1 et al, YEAR)"
The citation in the reference list: "Author NAME1, Author NAME2, ... (YEAR) article title. bioRxiv doi: XXX"
- in response to refs, please write in more circumspect manner - also crystal is not absolute truth

Use the link below to submit your revision:

Link Not Available

Referee #2:

Forker et al report the crystal structure of a short helical peptide of the RAD18 protein bound to the RAD18 interactor MAGEA4, an important regulator of RAD18's activity and stability.

This manuscript was submitted in response to a recently published EMBO J. paper (Griffith-Jones et al, 2024) focussing on the

same interaction but coming to a different conclusion about the orientation of the helix with respect to MAGEA4. While Forker et al used X-ray crystallography to determine the structure of the complex, Griffith-Jones et al had used AlphaFold modelling in combination with NMR studies and had supported their findings by solid mutagenesis and activity measurements. In the present manuscript, Forker et al recreate the erroneous AlphaFold model and come to the conclusion that in fact both orientations would be consistent with the NMR data in the original publication. In the response to the new manuscript, the corresponding author of the first paper admits the ambiguity of the NMR data but makes the point that their conclusions are unaffected by the orientation of the helical peptide, as the binding site on MAGE4A was correctly localized and pertinent mutations do interrupt the interaction and affect RAD18 activities and stability.

I consider the arguments brought forth from both sides as valid. Forker et al have clarified an AlphaFold-derived artefact and highlighted the importance of genuine structural research for elucidating molecular interactions. At the same time, Bhogaraju does not contest these findings but convincingly argues that the conclusions of Griffith-Jones still apply. I consider the manuscript and the response as justified and appropriate for publication because highlighting the correct orientation of the helix is important for attempts at developing inhibitors for this medically important interaction. I would recommend that Bhogaraju's response be shortened overall and possibly use a less defensive tone, which is not needed in this situation, given the validity of the original biochemical assays.

Referee #3:

The data reported in Griffith-Jones et al. are sound and appear to be consistent with both the AlphaFold model and the crystal structure. This raises the question as to which one is "right." The answer may be that both are "correct," meaning that both orientations exist in solution. What is observed in the crystal structure is an orientation that was able to crystallize; one should not infer that is the only form (or even the predominant one) in solution. The AlphaFold protocol used by Griffith-Jones et al. may not have calculated a sufficient number of models to reveal other possible orientations, including the one observed in the crystal. The use of a short (~30-40 residue) fragment of the binding partner, R6BD, may be a source of the apparent conflict, as it lacks to the larger molecular context that could serve to define a unique binding orientation between the two proteins.

Griffith-Jones et al. gathered NMR data for only one side of the interaction while the other side (R6BD) was "silent." Additional information from the R6BD side of the equation might have better defined the orientation. For example, use of a version of the R6BD with a spin-label attached to one or the other end would reveal whether R6BD binds in a single manner or has multiple types of interactions with MAGE. With reciprocal constraints on both sides, the authors could have used the well-vetted HADDOCK software to calculate models consistent with experimental results.

As it stands, the crystal structure shows a different orientation of a short region of Rad6 binding to MAGE4A that is consistent with the experimental results. It therefore should be published. That said, it would be wrong for readers to conclude that this structure is THE structure-an all-too-common inference. It is the structure of the complex that crystallized; neither set of authors provide evidence that defines the orientation(s) of the binding domain that exist in solution. I would encourage both sets of authors to be mindful of this reality; 1) the authors of the crystal structure should state this fact clearly and 2) the authors of the original paper should not be so quick to accept the crystal structure as "ground truth."

In summary, I recommend that both the manuscript reporting the crystal structure and the response from the original authors be published. Both should be encouraged to revise their pieces to reflect the remaining uncertainty. I believe that similar situations are likely to arise in the future. It would benefit the community and readership to be as clear as possible regarding the limitations of both predictive models and of experimental data. In this case, additional experimental data might have led to a different conclusion and, therefore, final model. The Editor might consider including an Editor's Comment or short opinion piece highlighting the pitfalls and limitations of using AI-generated models as well as the importance of keeping in mind that what is observed in a crystal or on a cryo-EM grid is representative of species in solution, but may not tell the full story.

Editor:

As you will see from their feedback copied below, they however suggest a certain amount of rephrasing for both pieces; in particular, the structural referee raises the well-taken point that also a crystal structure by itself does not necessarily reflect the physiological in-solution conformation.

SPECIFIC GUIDANCE:

- Please upload text and figure as separate, individual files (i.e., one file per figure). Please change Figures S1/2 into Expanded View Figures: calling them Figure EV1/2. Please retain the legends for both main and expanded view figures at the end of the main text. Lastly, please also upload the data collection table separately as "Table EV1", and adjust its callout accordingly.

- When referring to figures in the Griffith-Jones et al paper in the text, please try to differentiate them clearly from references to your own current figures, so that typesetters will not be confused and add inappropriate links

- After the method section, please include a "Data Availability" section (suggested wording: "The structural coordinates from this publication have been deposited to the [name of the database] database [URL] and assigned the identifier [accession #]."); and a Disclosure and Competing Interests Statement (for details, see <https://www.embopress.org/competing-interests>)

- Please adjust the format for citation of preprints as specified in our author guidelines:

The citation in the text should be: "(preprint: NAME1 et al, YEAR)"

The citation in the reference list: "Author NAME1, Author NAME2, ... (YEAR) article title. bioRxiv doi: XXX"

- in response to refs, please write in more circumspect manner - also crystal is not absolute truth

Response:

We have made the requested editorial changes about the Expanded View (EV) Figures and Table. We have revised the manuscript to state the potential limitation of the crystal structure in a new paragraph titled "*Limitation of the current study*".

Referee #3:

The data reported in Griffith-Jones et al. are sound and appear to be consistent with both the AlphaFold model and the crystal structure. This raises the question as to which one is "right." The answer may be that both are "correct," meaning that both orientations exist in solution. What is observed in the crystal structure is an orientation that was able to crystallize; one should not infer that is the only form (or even the predominant one) in solution. The AlphaFold protocol used by Griffith-Jones et al. may not have calculated a sufficient number of models to reveal other possible orientations, including the one observed in the crystal. The use of a short (~30-40 residue) fragment of the binding partner, R6BD, may be a source of the apparent conflict, as it lacks to the

larger molecular context that could serve to define a unique binding orientation between the two proteins.

Griffith-Jones et al. gathered NMR data for only one side of the interaction while the other side (R6BD) was "silent." Additional information from the R6BD side of the equation might have better defined the orientation. For example, use of a version of the R6BD with a spin-label attached to one or the other end would reveal whether R6BD binds in a single manner or has multiple types of interactions with MAGE. With reciprocal constraints on both sides, the authors could have used the well-vetted HADDOCK software to calculate models consistent with experimental results.

As it stands, the crystal structure shows a different orientation of a short region of Rad6 binding to MAGE4A that is consistent with the experimental results. It therefore should be published. That said, it would be wrong for readers to conclude that this structure is THE structure-an all-too-common inference. It is the structure of the complex that crystallized; neither set of authors provide evidence that defines the orientation(s) of the binding domain that exist in solution. I would encourage both sets of authors to be mindful of this reality; 1) the authors of the crystal structure should state this fact clearly and 2) the authors of the original paper should not be so quick to accept the crystal structure as "ground truth."

In summary, I recommend that both the manuscript reporting the crystal structure and the response from the original authors be published. Both should be encouraged to revise their pieces to reflect the remaining uncertainty. I believe that similar situations are likely to arise in the future. It would benefit the community and readership to be as clear as possible regarding the limitations of both predictive models and of experimental data. In this case, additional experimental data might have led to a different conclusion and, therefore, final model. The Editor might consider including an Editor's Comment or short opinion piece highlighting the pitfalls and limitations of using AI-generated models as well as the importance of keeping in mind that what is observed in a crystal or on a cryo-EM grid is representative of species in solution, but may not tell the full story.

Response:

We agree with the reviewer and have added a section titled "Limitation of the current study" in the revised manuscript to address these critiques.